# Design and Bench Testing of a Novel, Pediatric, Non-Invasive, Bubble Bilevel Positive Pressure Ventilation Device

**DOI:** 10.3390/bioengineering12070697

**Published:** 2025-06-26

**Authors:** Ibukun Sonaike, Robert M. DiBlasi, Jonathan Arthur Poli, Andrew Vamos, Ofer Yanay, Amelie von Saint Andre-von Arnim

**Affiliations:** 1Department of Pediatrics, Hennepin Healthcare, Minneapolis, MN 55415, USA; 2Global Pediatrics Program, Division of Pediatric Critical Care, Department of Pediatrics, University of Minnesota, Minneapolis, MN 55414, USA; 3Center for Respiratory Biology and Therapeutics, Seattle Children’s Research Institute, Seattle, WA 98101, USA; robert.diblasi@seattlechildrens.org; 4Respiratory Therapy Department, Seattle Children’s Hospital, Seattle, WA 98105, USA; 5Center for Integrative Brain Research, Seattle Children’s Research Institute, Seattle, WA 98101, USA; jonathan.poli@protonmail.com; 6Continuous Improvement and Innovation, Seattle Children’s Research Institute, Seattle, WA 98101, USA; andrew.c.vamos@gmail.com; 7Department of Pediatrics, Division of Pediatric Critical Care Medicine, University of Washington, Seattle Children’s Hospital, Seattle, WA 98105, USA; ofer.yanay@seattlechildrens.org (O.Y.); ameliev@uw.edu (A.v.S.A.-v.A.); 8Department of Global Health, University of Washington, Seattle, WA 98195, USA

**Keywords:** BiPAP, bubble CPAP, respiratory support, ventilator, pediatrics, low- and middle-income countries

## Abstract

Acute lower respiratory tract infections are a leading cause of death in individuals under the age of 5 years, mostly in low- and middle-income countries (LMICs). The lack of respiratory support systems contributes to the poor outcomes. Bubble CPAP is widely used for non-invasive respiratory support, but sicker children often require support over what CPAP provides in the form of BiPAP. We developed and tested a simple bubble-based bilevel ventilator (Bubble bi-vent) and compared it with a standard care BiPAP device. The bubble bilevel device consisted of a single tube submerged in a water-sealed column to maintain end-expiratory positive airway pressure. It moves vertically via an electric motor to also provide inspiratory positive airway pressure for augmentation of lung volumes, with the duration and frequency of breaths controlled by a microprocessor. We tested this novel device in passively breathing mechanical lung models for infants and small children. We compared pressure and tidal volume delivery between the novel device and a Trilogy BiPAP ventilator. The results showed that the Bubble bi-vent could deliver set pressures in a mechanical lung and was comparable to a standard Trilogy ventilator. While two different bubble-based bilevel pressure devices have been piloted for neonates and adults, our results demonstrate the feasibility of bubble bilevel ventilation for infants and small children with moderate to severe lung disease for whom this was previously not described.

## 1. Introduction

Acute lower respiratory tract infections (ALRTI) are one of the leading causes of death, resulting in an estimated 725,557 deaths in children under 5 years of age in 2021 [1]. Most of these ALRTI-related deaths occur in low- and middle-income countries (LMICs) [1]. While the causes for this staggering mortality from largely preventable ALRTI are complex, insufficient human resources and unavailability of oxygen therapy and respiratory support are important factors in LMICs [2,3,4,5,6]. Non-invasive ventilation (NIV) is the delivery of mechanical respiratory support without the need for endotracheal intubation through an interface (nasal or mask) that delivers continuous positive airway pressure (CPAP) or bilevel positive airway support (BiPAP) [7]. NIV offers the ability to reduce patients’ work of breathing and improve respiratory gas exchange while avoiding the risks and complications related to the placement of an endotracheal tube, administration of sedation, and delivery of invasive mechanical ventilation. However, in infants and small children, finding a well-fitting interface (nasal cannula or mask) for different age groups and the potential need for low-dose sedation to tolerate them can sometimes pose a challenge [8,9]. Given the high global burden of pediatric respiratory disease and the limited skills and capacity for invasive mechanical ventilation, there is justification for making low-cost, effective NIV devices more readily available in LMICs [3,6,10].

Bubble continuous positive airway pressure (bCPAP) is the most widely used NIV system for neonates and small infants in LMICs [3], requiring minimal training and technical skills for its operation while being low cost compared to mechanical ventilation [11,12]. bCPAP generates pressure by passing air through a tube submerged in water, creating pressure oscillations. Generated pressures are applied to a patient and help to promote airway patency, prevent alveolar collapse, and enhance gas exchange [13,14]. Evidence shows that bubbling improves gas exchange via the low-amplitude, high-frequency oscillations delivered to the alveoli [14]. There is good evidence for effective bCPAP use to support spontaneously breathing neonates and small infants. For children up to 5 years of age, randomized controlled trials of bCPAP therapy to treat ALRTI in LMICs have shown mixed results: one trial showed an 11% absolute mortality reduction compared with low-flow oxygen [13], a second resulted in a 4% absolute mortality reduction in infants but no significant difference in children aged 1–5 years [15], a third reported an absolute 6% increase in mortality with bCPAP [16], and a fourth showed a reduced risk of treatment failure and in-hospital mortality in children with bCPAP compared with the use of standard low-flow oxygen therapy [17].

However, infants and children with moderate to severe respiratory distress often cannot be supported sufficiently with conventional bCPAP. They require higher levels of NIV, such as BiPAP or invasive ventilation [18]. As the name implies, BiPAP delivers two set levels of positive airway pressure, one during inspiration (IPAP) and one during expiration (EPAP), augmenting lung volumes, reducing the work of breathing, and improving gas exchange [7]. This usually requires a mechanical ventilator or a dedicated BiPAP machine with proprietary single-limb breathing circuits and masks, which are expensive and not readily available in many LMIC settings [19]. A simple bilevel positive pressure prototype for pre- and full-term neonates based on the principle of bCPAP has been developed by John et al. [20,21,22], and another for adults has been developed by Poli et al. [23], both tested in vitro and the former also in vivo. Despite this advance, there is no widely available, affordable, and effective bubble-based bilevel NIV system for infants beyond the neonatal stage and children under 5 years of age, given the high burden of respiratory illness and the limitations of current technologies in LMICs. This is a significant gap to fill due to the persistently high global mortality rate from acute lower respiratory tract infections in children for whom this technology would serve.

The Bubble bi-vent studied here is a novel respiratory support apparatus based on the hydro-pneumatic principles of bCPAP using an underwater seal at a known, controllable depth to generate inspiratory and expiratory pressures with stochastic pressure oscillations. This study aimed to test this low-cost bubble bilevel positive pressure ventilator for use in pediatrics, spanning infants to toddlers and small children up to 20 kg. We tested the performance of the new Bubble bi-vent across a range of bias flow rates, pressure settings, lung compliance states, and airway interfaces in a mechanical lung model. We hypothesized that the delivery of IPAP and EPAP in healthy and ARDS mechanical lung models is similar when comparing the Bubble bi-vent to a commercially available BiPAP device.

## 2. Materials and Methods

This study followed an incremental model involving laboratory designing of the novel Bubble bi-vent and detailed bench testing in healthy and diseased lung models. The Bubble bi-vent was also compared to a conventional BiPAP ventilator.

### 2.1. Bubble Bi-Vent Design

The Bubble bi-vent consists of a dynamic expiratory tube that moves vertically within a water-sealed column via a custom jig with an electric-powered motor, the Pololu 4.4:1 Metal Gearmotor 25D × 63L mm HP 12V with 48 CPR Encoder (Pololu Robotics & Electronics, NV, USA) to provide bilevel pressures (Figure 1, also Figure A1 in Appendix A for scale). The tube and belt are mounted vertically using custom 3D-printed parts and pulleys so that any rotational motor movement results in linear actuation of the tube to variable submersion depths within the water column. The device motor uses encoder capabilities with closed-loop feedback to control precise vertical tube positioning beneath the water column to maintain accurate pressures (±0.5 cm of H_2_O) and limit errors from buoyancy and bubbling forces.

The motor rapidly fluctuates between two set points, the IPAP and EPAP, using a programmed microcontroller unit, the Cytron 10A Bi-Directional DC Motor Driver Shield for Arduino (Cytron Technologies, Penang, Malaysia). The settings for IPAP, EPAP, respiratory rate (RR), and inspiratory (T_I_) and expiratory (T_E_) times are adjustable on the microcontroller unit (MCU) via rotary dials and displayed on a liquid-crystal display (LCD) screen. The microcontroller unit (MCU) and peripherals are powered by a 5V USB connection (USBAB2MR), (StarTech, OH, USA) and the motor driver is powered by a 12V DC (GST60A12-P1J) source, (Mean Well, CA, USA) connected to wall power. 

The Bubble bi-vent is designed for use with a range of breathing gases, such as room air and oxygen, which are introduced via an inspiratory tubing (pediatric size) connected to an air source. Exhaled gases exit the system by bubbling out through the water column at two different pressure settings to generate IPAP and EPAP, which are transmitted to the patient. The Bubble bi-vent is designed to incorporate existing clinical devices such as flow meters, breathing gas blenders, a gas heater and humidifier, and a range of patient interfaces such as endotracheal tubes and bi-nasal prongs. The Bubble bi-vent in its current prototype does not synchronize with spontaneous breaths, and experiments were performed in passive lung models only.

### 2.2. Experimental Set-Up

We conducted four prospective in vitro studies to assess the performance of the Bubble bi-vent device in a 20 kg small-child model and 4 kg infant model using the ASL 5000 test lung (Ingmar Medical, Pittsburgh, PA, USA), a digitally controlled, high-fidelity breathing simulator that utilizes mathematical modeling to simulate size- and disease-specific pulmonary mechanics (Table 1) [24,25,26]. Delivered airway pressures were measured by the test lung.

Two 3D-printed upper anatomic airway models (infant and pediatric) were connected in series to the test lung (Figure 2) for non-invasive experiments [27,28]. The delivered pressures generated by the devices were measured within the test lung distal to the 3D airways. Non-invasive support interfaces were applied to the anatomic airway models via facemasks and nasal prongs (RAM, Neotech, Valencia, CA, USA). For invasive support testing, 4.5 mm- and 3.5 mm-sized endotracheal tubes (ETTs) for small-child and infant models were used, respectively. Direct connection involved directly connecting the ventilator to the ASL 5000 test lung without using an ETT or non-invasive interface.

### 2.3. Experimental Procedures

Test 1 involved measuring the Bubble bi-vent generated airway pressures in the lung model at bias flows of 5, 8, 10, and 15 L/min (L/min) to assess the effect of the gas source flow on the delivered airway pressures (Table 2). The Bubble bi-vent was set at IPAP/EPAP 12/6 cm H_2_O. The experiments were conducted in infant and small-child models using a direct connection to the mechanical lung to isolate the effects of different bias flows.

Test 2 investigated Bubble bi-vent pressure delivery in healthy and diseased lung models. Three different Bubble bi-vent pressure settings (IPAP/EPAP) were used: 12/6, 15/8, and 18/10 cm H_2_O. The delivered airway pressures were measured for the infant and small child in healthy and ARDS test lung compliance and resistance states. The procedures were performed via direct connection of the ventilator to the test lung to isolate the effects of the different compliance and resistance states.

The third test evaluated the effects of different interfaces on Bubble bi-vent airway pressure delivery in healthy infant and small-child models. Bubble bi-vent pressure settings of IPAP/EPAP 15/8 cm H_2_O were used with the following interfaces: direct connection, endotracheal tube, face mask (small child), and nasal cannula (infant).

Test 4 compared airway pressure and tidal volume delivery between the Bubble bi-vent and the Trilogy ventilator (Respironics Trilogy 202, Philips, Amsterdam, The Netherlands). The latter is a commercially available subacute care ventilator with a turbine-based flow BiPAP function for adults and pediatric patients. The Trilogy ventilator employs a single limb circuit with a fixed leak exhalation valve with functionality for subject triggering, flow, and pressure measurements. It was used in pediatric BiPAP mode for this test. In test 4, a face mask applied to a 3D pediatric upper airway model was used at the following IPAP/EPAP settings for each ventilator: 12/6, 15/8, and 20/10 cmH_2_O. The comparative experiments were only performed for the small-child model and not for the infant model, as the Trilogy ventilator is designed to be used for patients > 5 kg.

The lung model parameters with associated pulmonary mechanics and size-specific interfaces are referenced in Table 1.

### 2.4. Data Analysis

The sampled data for each run of 20 breaths was extracted from the ASL 5000 test lung software and saved to a spreadsheet (Excel, Microsoft, Redmond, WA, USA). Descriptive statistics, including mean and standard deviation airway pressures, were calculated using MATLAB Data Analysis package (MathWorks, MA, USA, https://www.mathworks.com/products/matlab/data-analysis.html, accessed on 15 June 2025). Bubble bi-vent and Trilogy ventilator performance was compared using the set device pressure as the reference for measured test lung pulmonary pressures based on % Error calculation, where Error = [Measured Pressure − Set Pressure]/Set Pressure × 100.

## 3. Results

### 3.1. Test 1—Bias Flow Titration

Waveforms depicting airway pressure (with evidence of continuous oscillations during breath cycles) and tidal volume delivery are shown in Figure 3a and Figure 3b respectively. At Bubble bi-vent IPAP/EPAP pressure settings of 12/6 cmH_2_O, bias flows of 5 L/min provided mean pressure measurements in the healthy test lung of IPAP 12.2 (SD ± 0.18) and EPAP 6.4 (SD ± 0.40), and IPAP 13.7 (SD ± 0.16) and EPAP 5.9 (SD ± 0.32) in the small-child and infant models, respectively (Figure 4). Using higher bias flows led to increased measured IPAPs; at a bias flow of 10 L/min, IPAP/EPAP was 14 (SD ± 0.13)/6.5 (SD ± 0.41) in the small-child model and 15.5 (SD ± 0.19)/6 (SD ± 0.50) in the infant model. At bias flows of 15 L/min, mean pressures were as high as IPAP/EPAP of 14.9 (SD ± 0.3)/6.3 (SD ± 0.50) in the small-child model and 17.1 (SD ± 0.85)/6.2 (SD ± 0.52) in the infant model. For all following experiments, we used the minimum bias flows that produced consistent bubbling, which were 10 L/min in the small-child model and 5 L/min in the infant model.

### 3.2. Test 2—Bubble Bi-Vent Testing in Healthy and Sick Lungs

Measuring Bubble bi-vent delivered pressures at IPAP/EPAP settings of 12/6 cmH_2_O yielded a mean IPAP/EPAP of 13.6/6.1 and 15.1/6.0 in the healthy and diseased small-child lung models, respectively. Higher measured airway pressures were consistently observed in sick lungs at multiple pressure settings for both infant and small-child models (Table 3a,b).

### 3.3. Test 3—Bubble Bi-Vent Testing Across Different Interfaces

Different interfaces were used with the Bubble bi-vent, and the delivered airway pressures were measured in the small-child and infant lung models in both healthy and disease states (Figure 4). With device settings of 15/8 cmH_2_O in the healthy small-child lung model, the measured IPAP/EPAP was 16.3 (±0.07)/8.2 (±0.41) for direct connection, 15.6 (±0.12)/7.9 (±0.35) for ETT, and 13.7 (±0.05)/7.8 (±0.20) for face mask interfaces (Figure 5). The same device settings in the healthy infant model yielded a measured IPAP/EPAP of 17.1 (±0.10)/8.5 (±0.34) with direct connection, 16.4 (±0.07)/8.5 (±0.21) with ETT, and 14.9 (±0.08)/8.4 (±0.25) with nasal cannula applied to a 3D-printed infant airway model.

### 3.4. Test 4—Bubble Bi-Vent Comparison with Trilogy Ventilator

Bubble bi-vent performance in airway pressure delivery was compared to the Trilogy ventilator at three different pressure settings in healthy and diseased lungs with a non-invasive mask for each ventilator (Table 4a,b). Bubble bi-vent percent error measurements of the mean pressures ranged between 0 and 5% for IPAP and 7.5 and 12% for EPAP in the healthy lung model, respectively, compared to the mean percent error measurements of −11 to −8% for IPAP and −3 to −4% for EPAP for the Trilogy ventilator. In the pediatric ARDS model, the Bubble bi-vent percent error ranged between 1.3 and 3% for IPAP and 4 and 5% for EPAP. For the Trilogy ventilator, error measurements were −8 to −6.6% for IPAP and −5% for EPAP, respectively. Tidal volumes increased with increased delta pressures and were reduced in the ARDS compared to the health lung model for both the Bubble bi-vent and Trilogy ventilator.

## 4. Discussion

This pilot study demonstrated that the novel, simple Bubble bi-vent offers hydro-pneumatic respiratory support for both infants and children, comparable to standard BiPAP therapy with a conventional ventilator in mechanical lung models ranging from healthy to ARDS lungs. This could be a first step towards a potential new support option for young children with respiratory distress and failure in low-resource settings where invasive mechanical ventilation is frequently not available and bCPAP is not sufficient.

For test 1, bias flow is the continuous flow of air delivered through the system, and it is critical in generating the desired small-amplitude, high-frequency pressure oscillations, which in bCPAP have been shown to be beneficial in opening collapsed alveoli, as well as enhancing lung volume and gas exchange in preterm infants [14,20,29]. Higher amplitude pressure oscillations that were intermittently observed can be avoided using a diffuser (plastic barrier with small openings) [23,30], which requires follow-up experiments. We observed higher IPAPs and pressure fluctuations in the infant lung model with up-titration of bias flows, likely from increased airway resistance in less compliant and infant lungs. A lower bias flow rate was, therefore, chosen in our follow-up infant experiments. In test 2, in which the Bubble bi-vent was tested via direct connection, the measured airway pressures were higher than the set pressures, especially in lower compliance states and in the infant lung model. This is not surprising given our understanding of the effects of compliance in a closed system as well as increased airway resistance in smaller infants. There could also be a contribution of “noise” introduced by buoyancy and bubbling in the Bubble bi-vent system, which requires further evaluation in in vivo models. Non-invasive interfaces (mask and nasal cannula) led to lower airway pressures when compared to invasive delivery in test 3 (Figure 4), secondary to leaks introduced to the system via non-invasive interfaces. Hence, for the intended non-invasive use of the Bubble bi-vent, there is less concern about barotrauma from bubble BiPAP, although pneumothoraces have been described in bubble CPAP studies. While in invasive ventilation, the airway is sealed; in NIV, leaks of varying degrees occur, depending on the interface. Leaks during NIV can be a significant contributor to patient–ventilator asynchrony and compromise inspiratory and expiratory pressures and tidal volume delivery [31]. The most common cause of leak-related asynchrony during NIV is auto-triggering. Since the current Bubble bi-vent prototype does not have a trigger system, auto-triggering is less of a concern. Like other bubble respiratory devices, the Bubble bi-vent does not have leak compensatory functions. Clinical monitoring and frequent reassessments, titrating respiratory support to the patient’s clinical condition while ensuring the interfaces fit well, and limiting leaks will continue to be important when patients are supported with these forms of ventilatory support [32,33]. When comparing the delivered airway pressures between the Bubble bi-vent and standard Trilogy, the percent error range was comparable between the two devices. This is promising, given the need for further simple, low-cost, novel respiratory support devices for low-resource settings. The authors are not aware of generally accepted standards for precision deviations from set ventilator settings. While precise delivery of set ventilator variables is the goal, substantial deviations from targeted volume and pressure settings have been reported in clinical ventilator performance settings [34,35,36]. In vivo feasibility and safety studies will be needed to assess these factors and uncover other potential problems.

The recent COVID-19 global pandemic led to an estimated 18 million excess deaths globally [37] and has further highlighted the critical supply shortages of ventilators, even in well-resourced nations [38,39], making the need for additional affordable and simple-to-use ventilation technologies very obvious.

Respiratory technologies similar to the Bubble bi-vent for use in low-resource settings are the NeoVent, designed for the neonatal and premature infant population, and the Bubble-Vent for adults [23]. All three devices utilize hydro-pneumatics through an underwater seal at a known, controllable depth to generate inspiratory and expiratory pressures with stochastic pressure oscillations. The NeoVent uses a basket sleeve mechanism, the Bubble-Vent uses a 3D-printed float-valve system connected to two separate water baths of differing levels, and the Bubble bi-vent uses an electric motor for generating EPAP and IPAP [20,21,22,23]. Similar pressure waveform and tidal volume delivery was observed in the NeoVent and the Bubble bi-vent. The advantages of the Bubble bi-vent, when compared with the NeoVent, are the ability to precisely control the respiratory rate and inspiratory time as well as the capacity to support both infants and larger children. However, the NeoVent can run without electricity, which is a significant advantage in resource-poor settings. A rechargeable battery-driven motor set-up for the Bubble bi-vent could be a reasonable alternative for settings with limited electricity and should be evaluated in the future. In in-vivo studies, NeoVent bubble bilevel ventilation provided effective gas exchange in anesthetized, intubated young rabbits. The Bubble-Vent by Poli et al. [23] was able to provide timed, pressure-limited support in an adult mechanical lung model, but pressure delivery was less precise and with less volume delivery when compared to the Draeger VN500, a critical care ventilator [23]. The bias flow rates required for the Bubble-Vent, which range between 40 and 80 L/min, would be prohibitive in many resource-poor settings where medical air and oxygen supplies are often unavailable. No in vivo studies have been conducted using this device. None of the three bubble BiPAP devices can provide synchronized respiratory support. While synchronization will allow for better tolerance and optimal delivery of the intended support, a recent meta-analysis of BiPAP use in neonates reported reduced respiratory failure rates with both synchronous and non-synchronous support modalities [40]. None of the three bubble ventilators have inbuilt alarm and monitoring systems. If a patient on nasal bubble BiPAP stops breathing, opens their mouth, or the nasal interface is displaced or obstructed, bubbling will cease. Therefore, close monitoring of any bubble BiPAP device performance is essential. While bubble CPAP devices in LMICs currently lack alarm systems [32], a simplified version of the monitoring system employed by Menniti et al. [41] might serve Bubble bi-vent in the future.

While the Bubble bi-vent test results are promising, clinical relevance is limited by the passive mechanical lung model. Hence, its effects on an actively breathing child cannot be estimated. This may impact its clinical benefits and will need to be evaluated in future animal model studies. Appropriately fitting non-invasive interfaces can be a challenge to find, especially in low-resource settings, given the need for various sizes and shapes in children of multiple ages. This study used only two commercially available sizes of tested interfaces. The need for airflow rates of up to 10 L/min in pediatric patients and constant electricity may limit the usability of this device in its current state in low-resource settings where oxygen and medical air availability may be restricted and power outages are common.

Ventilator device failure modes can be broadly categorized into mechanical failures, electrical failures, and user errors. This device would, thus, also require alarm systems and basic monitoring features, none of which were feasible to add to this limited proof-of-concept study. Mechanical failures for the Bubble bi-vent include motor failures, belt uncoupling with tube displacement, dislodgement of the water bath, and circuit leaks. The risks of motor, belt system, and tube dislodgement failures can be minimized by iterative design processes and rigorous testing. Versaci et al. [42] describe a novel technique using soft computing to classify structural defects in carbon fiber-reinforced polymer (CFPR) in biomedical devices. This could be used to assess the structural integrity of the 3D-printed parts of this novel ventilator. Electrical device failures include interruption of the power supply, for which, as mentioned, an integrated backup battery should be included. User errors include incorrect settings and training. The simplicity of the device will make errors due to incorrect settings less likely, but given the limited number and training of healthcare staff in many LMICs, adequate training and retraining on device operation and troubleshooting have been associated with survival for bubble CPAP in Sub-Saharan Africa and are essential [15,43]. The Bubble bi-vent was left to run reliably for up to 6 h, but further testing, including feasibility, safety, and durability, is necessary.

## 5. Conclusions

The results of this pilot study provide a proof-of-concept of the simple, novel Bubble bi-vent device in a passive lung model. Its pressure delivery in a non-invasive support mode is comparable to a conventional microprocessor-controlled subacute ventilator. It produces the desired small-amplitude, high-frequency pressure oscillations with known beneficial effects on lung recruitment and gas exchange. Used with the intended non-invasive interfaces, airway pressure delivery is consistent with set pressures. These results provide the first step towards a new respiratory support option for young children in low-resource settings who are disproportionately affected by high global mortality rates from acute respiratory illnesses. Further lung model feasibility studies will be needed to address safety and durability, while in vivo device testing is required in order to measure the precision of pressure delivery; physiological effects, including oxygenation and ventilation; and the possible clinical impact of the high-frequency pressure oscillations due to bubbling.

## Figures and Tables

**Figure 1 bioengineering-12-00697-f001:**
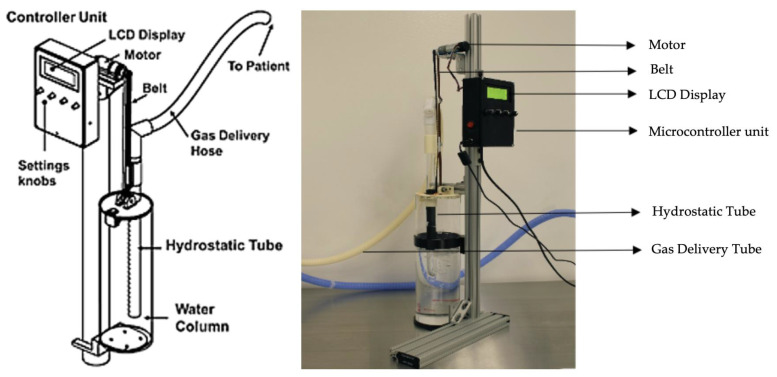
Bubble bi-vent device and motor parts. Device consists of a motor coupled with a hydrostatic mobile tube submerged in a water bath. The hydrostatic tube is connected to delivery tubing. See Table A1 in Appendix A for bill of materials.

**Figure 2 bioengineering-12-00697-f002:**
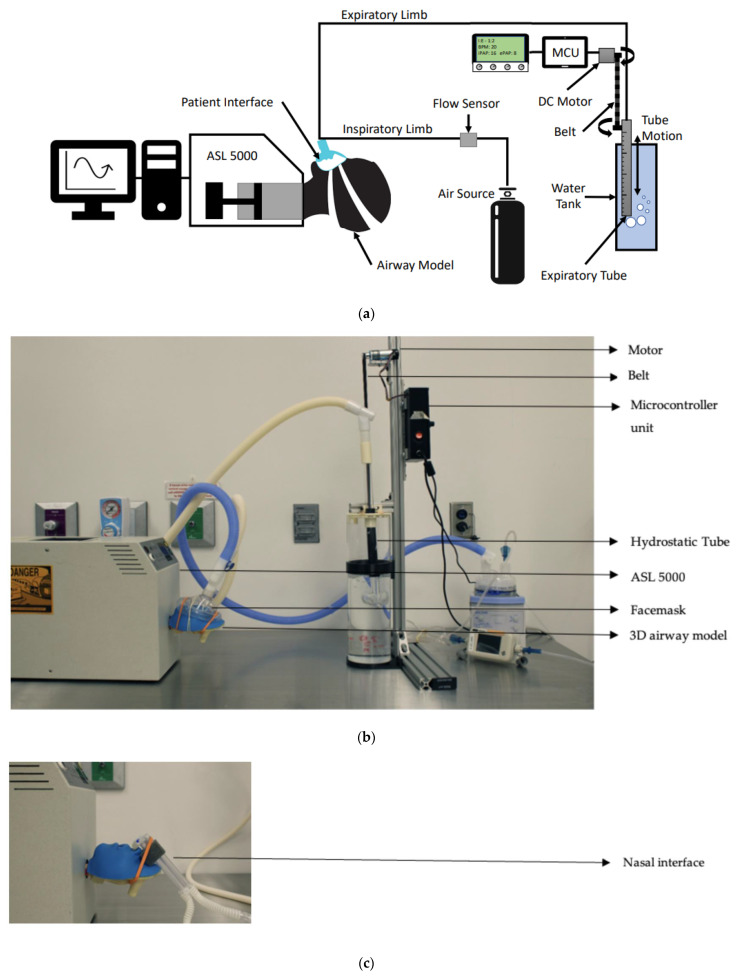
(**a**) Bubble bi-vent device and schematic experimental set-up with face mask interfaces. (**b**) Bubble bi-vent device connected to test lung using nasal canula interface. (**c**) Bubble bi-vent device connected to test lung using nasal interface.

**Figure 3 bioengineering-12-00697-f003:**
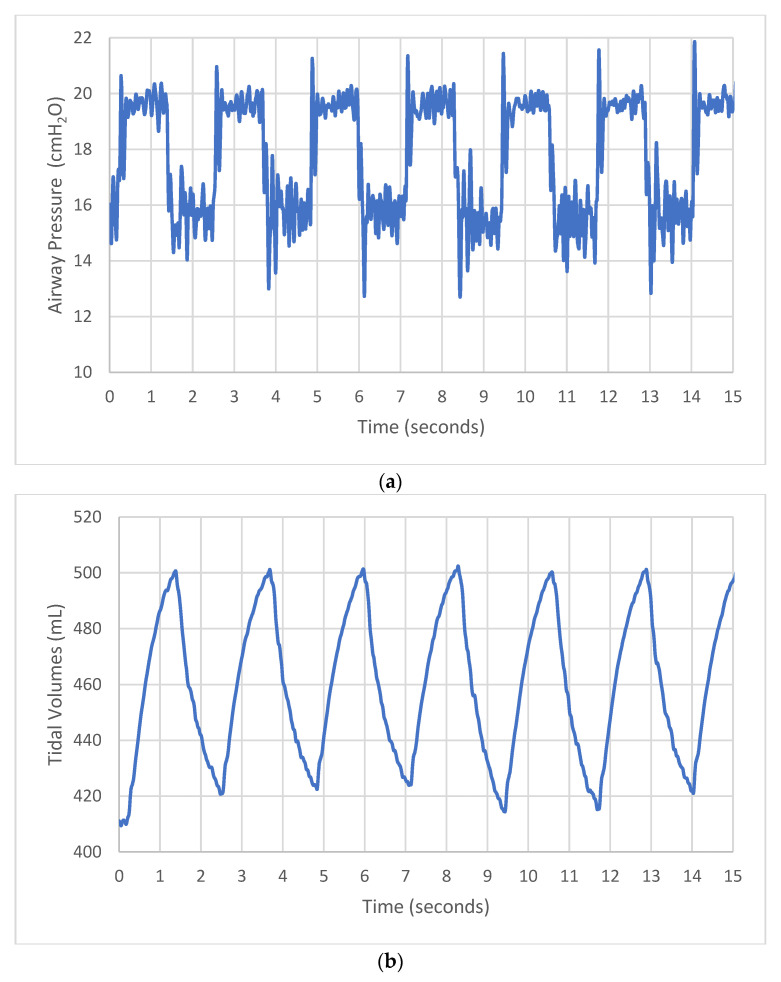
Bubble bi-vent pressure (**a**) airway pressure waveform (with evidence of continuous bubbling); (**b**) tidal volume waveforms at bias flows of 8 L per minute.

**Figure 4 bioengineering-12-00697-f004:**
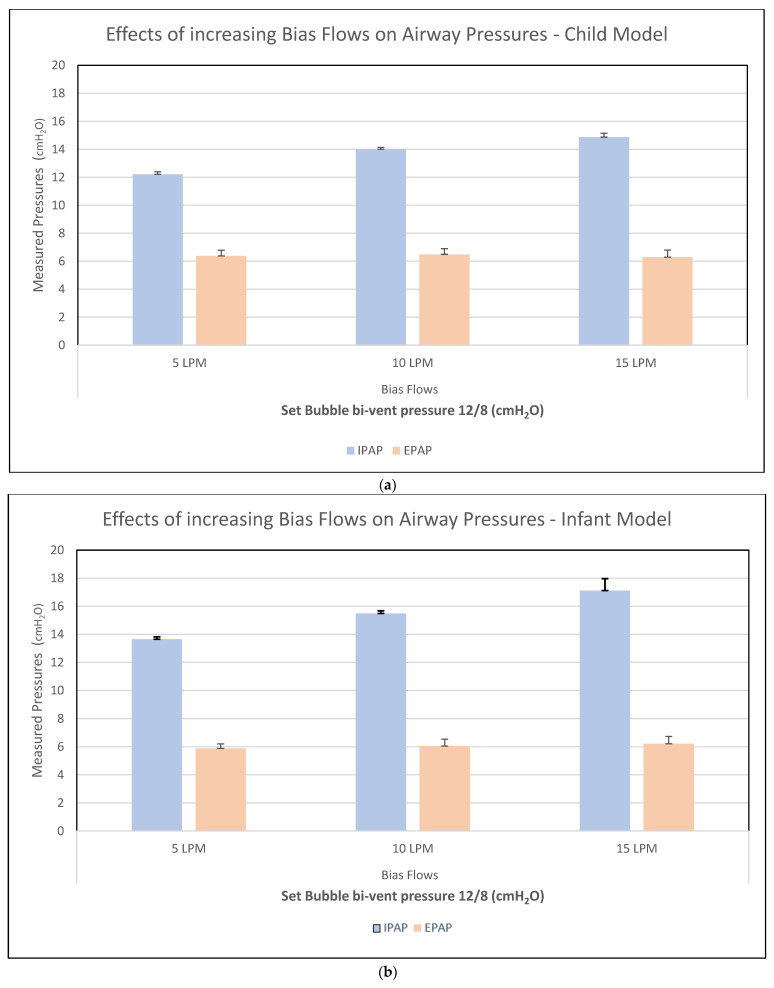
Measured airway pressures with increasing bias flows in healthy (**a**) small-child model and (**b**) infant model, respectively.

**Figure 5 bioengineering-12-00697-f005:**
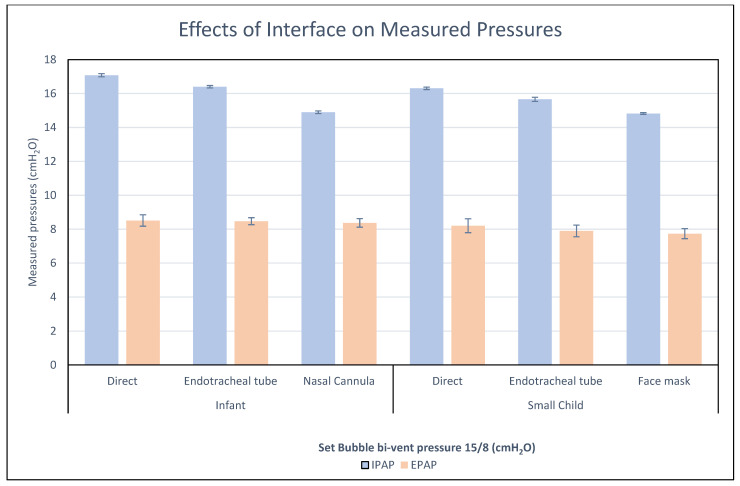
Measured pressures in healthy infant and small-child lung models for Bubble bi-vent used with different interfaces.

**Table 1 bioengineering-12-00697-t001:** Test lung model parameters with associated pulmonary mechanics and size-specific interfaces [24,25,26].

Parameters	Infant 4 kg Healthy Model	Infant 4 kg ARDS Model	Small Child 20 kg Healthy Model	Small Child 20 kg ARDS Model
Compliance (mL/cm H_2_O)	8	2	25	14
Resistance (cm H_2_O/L/s)	50	100	20	40
Non-invasive interface	Ram Nasal Cannula	Ram Nasal Cannula	Nonny Full Face Pediatric Mask	Nonny Full Face Pediatric Mask
Invasive interface	ETT 3.5 mm	ETT 3.5 mm	ETT 4.5 mm	ETT 4.5 mm
Without interface	Direct	Direct	Direct	Direct

ARDS = acute respiratory distress syndrome. ETT = endotracheal tube. Direct connection = connecting Bubble bi-vent to ASL test lung without the use of a mask, nasal cannula, or endotracheal tube interface.

**Table 2 bioengineering-12-00697-t002:** Experimental procedures.

**Experimental Procedures**	**Variable**		** 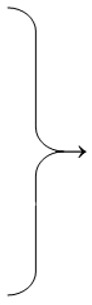 **	Measured IPAP and EPAP in mechanical lung
Test 1	Bias flow	5 10 15 20		
Test 2	Lung compliance and resistance	Healthy lungs ARDS lungs		
Test 3	Interface	Non-invasive Invasive Direct connection		
Test 4	Device type	Trilogy ventilator Bubble bi-vent		

Bias flow (liters/minute). Lung compliance (mL/cm H_2_O). Airway resistance (cm H_2_O/L/s).

**Table 3 bioengineering-12-00697-t003:** Effects of healthy versus ARDS lung compliance states on Bubble bi-vent delivered mean airway pressures with direct connection of ventilator to (**a**) small-child model and (**b**) infant lung model.

(a)					
Vent Parameters		Healthy		ARDS	
Set IPAP (cm H_2_O)	Set EPAP (cm H_2_O)	Measured IPAP (cm H_2_O)	Measured EPAP (cm H_2_O)	Measured IPAP (cm H_2_O)	Measured EPAP (cm H_2_O)
12	6	13.6 ± 0.16	6.1 ± 0.55	15.1 ± 0.46	6.0 ± 0.39
15	8	16.5 ± 0.16	8.2 ± 0.41	18.1 ± 0.15	8.2 ± 0.29
18	10	18.4 ± 0.11	9.1 ± 0.46	20.2 ± 0.19	10.62 ± 1.82
**(b)**					
**Vent Parameters**		**Healthy**		**ARDS**	
**Set IPAP ** **(cm H_2_O)**	**Set EPAP ** **(cm H_2_O)**	**Measured IPAP (cm H_2_O)**	**Measured EPAP (cm H_2_O)**	**Measured IPAP (cm H_2_O)**	**Measured EPAP (cm H_2_O)**
12	6	14.2 ± 0.16	6.6 ± 0.32	16.2 ± 0.31	6.3 ± 0.52
15	8	17.2 ± 0.09	8.4 ± 0.26	19.8 ± 0.22	8.5 ± 0.57
18	10	20.2 ± 0.04	10.6 ± 0.32	22.2 ± 0.21	10.6 ± 0.48

IPAP = inspiratory positive airway pressure. EPAP = expiratory positive airway pressure. ARDS = acute respiratory distress syndrome.

**Table 4 bioengineering-12-00697-t004:** Comparison of non-invasive Bubble bi-vent airway pressure delivery with conventional Trilogy ventilator in (**a**) small-child with healthy lung and (**b**) small-child with ARDS models.

(a)								
Ventilator Type	Set IPAP (cmH_2_O)	Set EPAP (cmH_2_O)	Measured IPAP (cmH_2_O)	IPAP Error %	Measured EPAP (cmH_2_O)	EPAP Error %	Tidal Volume (mL)	Tidal Volume (mL/kg)
Bubble bi-vent	12	6	12.6 ± 0.03	5	6.7 ± 0.28	12	104.6 ± 1.34	5.2
	15	8	15.5 ± 0.04	3	8.6 ± 0.26	7.5	119.6 ± 1.71	5.9
	20	10	20 ± 0.06	0	10.8 ± 0.45	8	157.1 ± 2.29	7.9
Respironics Trilogy 202	12	6	11 ± 0.02	−8	5.8 ± 0.01	−3	95.7 ± 0.19	4.8
	15	8	**13.6** ± 0.02	−9	**7.7** ± 0.02	−3	109.0 ± 0.15	5.5
	20	10	**17.9** ± 0.03	−11	**9.6** ± 0.02	−4	151.5 ± 0.11	7.6
**(b)**								
**Ventilator Type**	**Set IPAP (cmH_2_O)**	**Set EPAP (cmH_2_O)**	**Measured IPAP (cmH_2_O)**	**IPAP Error %**	**Measured EPAP (cmH_2_O)**	**EPAP Error %**	**Tidal Volume (mL)**	**Tidal Volume** **(mL/kg)**
Bubble bi-vent	12	6	12.2 ± 0.04	1.6	6.3 ± 0.18	5	60.9 ± 0.54	3.0
	15	8	15.2 ± 0.04	1.3	8.4 ± 0.27	5	71.1 ± 0.63	3.6
	20	10	20.6 ± 0.05	3	10.4 ± 0.17	4	104.2 ± 0.95	5.2
Respironics Trilogy 202	12	6	11.2 ± 0.02	−6.6	5.7 ± 0.01	−5	61.8 ± 0.08	3.1
	15	8	13.9 ± 0.03	−7.3	7.6 ± 0.02	−5	70.7 ± 0.08	3.5
	20	10	18.4 ± 0.02	−8	9.5 ± 0.02	−5	100 ± 0.16	5.0

Percent Error = (Measured Pressure − Set Pressure)/Set Pressure × 100. IPAP = inspiratory positive airway pressure. EPAP = expiratory positive airway pressure. ARDS = acute respiratory distress syndrome.

## Data Availability

Please reach out to the corresponding author for data availability.

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
