# Peer review of "Design and Bench Testing of a Novel, Pediatric, Non-Invasive, Bubble Bilevel Positive Pressure Ventilation Device"

_bioengineering, 2025, doi:10.3390/bioengineering12070697_

Round 1
Reviewer 1 Report
Comments and Suggestions for Authors
This paper presents a new two-level non-invasive bubble ventilation device for paediatric use. The study addresses an important challenge because it targets low- and middle-income countries (LMIC), where access to advanced ventilatory support is limited. The performance of the device was compared with a standard BiPAP ventilator in various mechanical lung models. The document is well-structured and clearly written, and it addresses a relevant problem with an innovative and low-cost solution. However, several aspects need clarification and improvement before the paper can be considered for publication.
1) The novelty of the device is well described, but the distinction between this device and other models should be better emphasised in the Introduction. In this regard, I suggest that the authors integrate the study and include this study doi:10.3390/electronics13040790 in the bibliography, as it addresses the issue of continuous monitoring and patient-device interaction in home environments. Moreover, it proposes an integrated ventilation and monitoring system that can serve as a reference to improve the section on the clinical limitations of the Bubble bi-vent, which currently *does not include any form of sensing or synchronisation with the patient's respiratory effort. Finally, the authors should explain in detail what the technical or clinical advantages of the proposed model are compared to others?
2) The description of the experimental procedures is generally adequate, but some steps require a more explicit justification. The lack of synchronisation with spontaneous breaths is a known limitation, but it should be acknowledged earlier in the Methods section as a key factor affecting the translation potential.
3) Although the data on pressure and tidal volume are presented comprehensively, the clinical relevance of the percentage error margins (e.g., 5–12%) is not discussed. Would this deviation be acceptable in clinical use? The variability of data under different interfaces, for example direct vs. nasal cannula, is critical. Can the authors comment on the compensation of losses or the optimisation of the interface?
4) The discussion thoroughly compares previous devices, but it could benefit from a paragraph dedicated to potential failure modes (e.g., motor jamming, tube displacement) and how they can be mitigated. The statement regarding the "proof-of-concept" is correct, but the conclusion should emphasise more strongly that usability studies are essential steps before translational application. Regarding potential failure modes, a study that the authors should integrate into the paper is doi:10.2478/jee-2025-0007. Although not directly related to ventilation, this study demonstrates the application of soft computing methods for the non-invasive detection of structural anomalies in materials, particularly in biomedical devices such as the non-invasive and portable ventilator.
5) The authors should improve Figure 1 with clearer labels for the components and the inclusion of the scale. Tables 3 and 4 present important results but could be more comprehensible if reorganised with bold headings for the key results.
6) Line 86–89. It would be useful to specify the precise age/weight range that the device is designed to cover.
7) Line 278–280: This statement should be softened or supported with stronger contextual evidence “first step towards a new support option…”.
8) Typographical errors: A few instances of inconsistent spacing and punctuation, particularly in reference formatting.
Author Response
General response
We sincerely thank you for your constructive comments, especially as regards suggestions for improving the description of the device, including some of the technical limitations and discussion around future improvements. Find below comments and corresponding revisions addressing raised concerns. We believe the quality of the manuscript is stronger having addressed your comments.
Comments and Suggestions for Authors from Reviewer 1
This paper presents a new two-level non-invasive bubble ventilation device for paediatric use. The study addresses an important challenge because it targets low- and middle-income countries (LMIC), where access to advanced ventilatory support is limited. The performance of the device was compared with a standard BiPAP ventilator in various mechanical lung models. The document is well-structured and clearly written, and it addresses a relevant problem with an innovative and low-cost solution. However, several aspects need clarification and improvement before the paper can be considered for publication.
Comment 1
The novelty of the device is well described, but the distinction between this device and other models should be better emphasised in the Introduction. In this regard, I suggest that the authors integrate the study and include this study doi:10.3390/electronics13040790 in the bibliography, as it addresses the issue of continuous monitoring and patient-device interaction in home environments. Moreover, it proposes an integrated ventilation and monitoring system that can serve as a reference to improve the section on the clinical limitations of the Bubble bi-vent, which currently *does not include any form of sensing or synchronisation with the patient's respiratory effort. Finally, the authors should explain in detail what the technical or clinical advantages of the proposed model are compared to others?
Response 1
Thank you for this helpful feedback. We have added the reference suggested by reviewer 1 in Discussion section now Line 383. In addition, we have added a more detailed discussion on synchronization and alarm monitoring systems to the Discussion section from line 378, also see line 395.
In terms of technical and clinical advantages of the Bubble-bi-vent compared to other devices, we expanded the existing discussion (from line 353) to more clearly state difference and similarities between them.
Comment 2
The description of the experimental procedures is generally adequate, but some steps require a more explicit justification. The lack of synchronization with spontaneous breaths is a known limitation, but it should be acknowledged earlier in the Methods section as a key factor affecting the translation potential.
Response 2
The lack of synchronization with spontaneous breaths is now being acknowledged earlier in Methods now line 128-130.
Comment 3
Although the data on pressure and tidal volume are presented comprehensively, the clinical relevance of the percentage error margins (e.g., 5–12%) is not discussed. Would this deviation be acceptable in clinical use? The variability of data under different interfaces, for example direct vs. nasal cannula, is critical. Can the authors comment on the compensation of losses or the optimisation of the interface?
Response 3
Thank you for bringing up this important point on acceptable deviation of respiratory support delivery. Acceptable precision standards of respiratory support delivery are device dependent. The authors are not aware of general ISO standards for acceptable deviation ranges for invasive and non-invasive respiratory pressure delivery. We did add a discussion of the recent literature regarding deviations in pressure and volume delivery found in clinical practice which does, of course, not mean that these deviations are acceptable.
The variability of pressure delivery between direct connection, i.e. a sealed system, and non-invasive support with leaks is expected and well described in the literature. Leaks can lead to decreased efficacy of NIV and patient-ventilator asynchrony, frequently due to auto-triggering. As the Bubble-bi-vent in its current prototype does not include a trigger or sensing system, auto-triggering is less of a concern. However, decreased pressure delivery due to leaks remains a concern. As described for the implementation of bubble CPAP, clinical monitoring and frequent reassessments, titrating respiratory support to patient’s clinical condition while ensuring interfaces fit well to limit leaks will continue to be important.
We added these important points to the discussion (line 326-348)
Comment 4
The discussion thoroughly compares previous devices, but it could benefit from a paragraph dedicated to potential failure modes (e.g., motor jamming, tube displacement) and how they can be mitigated. The statement regarding the "proof-of-concept" is correct, but the conclusion should emphasise more strongly that usability studies are essential steps before translational application. Regarding potential failure modes, a study that the authors should integrate into the paper is doi:10.2478/jee-2025-0007. Although not directly related to ventilation, this study demonstrates the application of soft computing methods for the non-invasive detection of structural anomalies in materials, particularly in biomedical devices such as the non-invasive and portable ventilator.
Response 4
Thank you for this important suggestion. We have added a paragraph on ventilator failures modes (from line 395) with inclusion of the suggested reference to the discussion (line 400), as well as the need for usability studies to the conclusion.
Comment 5
The authors should improve Figure 1 with clearer labels for the components and the inclusion of the scale. Tables 3 and 4 present important results but could be more comprehensible if reorganised with bold headings for the key results.
Response 5
The clarity of Figure 1 has been improved. We have highlighted the key results in the tables.
Comment 6
Line 86–89. It would be useful to specify the precise age/weight range that the device is designed to cover.
Response 6
Thank you. This has now been specified now line 93-94.
Comment 7
Line 278–280: This statement should be softened or supported with stronger contextual evidence “first step towards a new support option…”.
Response 7
Statement has been modified to address remark.
Comment 8
Typographical errors: A few instances of inconsistent spacing and punctuation, particularly in reference formatting.
Response 8
The manuscript was reviewed and corrections made for typographical errors, spacing and punctuation inconsistencies and errors. References were reviewed as well.

Reviewer 2 Report
Comments and Suggestions for Authors
The rich content of this manuscript identifies a gap in non-invasive ventilation for the general category of infants beyond the neonatal period who may also be considered small children or toddlers. The research is very clearly outlined and the technology development, testing, and discussion is quite complete without any obvious deficiencies.
[Line 22-23] The reader is potentially left “hanging” on exactly the issue in the sentence “….but is insufficient for sicker children” Because? Due to? This, because the following sentence implies a solution to a problem that is ambiguous.
[Line 51-52] The introduction is absent further elaboration of “….finding a well-fitting interface for different age groups.” until you get to Table 1, where specific nasal cannula and masks are identified. It would help to add just a few words on what is meant by interface, as some readers may be thinking in terms of control system interface. This is elucidated in Lines 118-120, much farther into the manuscript, and then again in Lines 281-284.
[Line 56-57] The use of the reference seems to be more to emphasize the LMIC need versus explain the unique aspects of the bubble CPAP. Even just adding a sentence that with bCPAP bubbles formed in water create pressure oscillations to prevent alveolar collapse. As th reader moves forward, the distinction between the common CPAP and BiPap in the previous paragraph (lines 47-48) is lost in translation to the bCPAP comments. While not a big deal, you are eventually comparing and contrasting CPAP, BiPAP, bCPAP, and now bi-level-bCPAP and the reader is required to read and re-read to understand the nuances between them. A table to compare and contrast might be helpful.
[Lines 68-80] Excellent overall narrative leading to identification of the gap in technology, but is the issue about LMIC device cost, functionality, or both? It seems the current work offers the clinician a pathway where cost and functionality are not mutually exclusive. Perhaps the authors could consider adding a sentence that underscores the significance of the work in a more compelling way, such as: “There is no widely available, affordable, and effective bubble-based bilevel NIV system for infants beyond the neonatal stage and children under 5, despite the high burden of respiratory illness in this age group and the limitations of current technologies in LMICs.” If such a statement is true about the manuscript, it might help the reader for this type of direct statement to be added.
[Lines 98-102] Given the idea the device would meet the needs of LMIC, it may help to have in an appendix a summary of the bill-of-materials and cost comparison to traditional devices to put “low cost” in context. Indeed, the manuscript is centered on a pilot trial of an experimental set-up, but the pathway or motivation to translation would be enhanced. It is unclear if the need to have high-end technology “….using custom 3D-printed parts and pulleys…” (Line 102) would be required at the LMIC site.
The methods, procedures, results and statistics meet contemporary standards for clarity and quality. And the results do support the conclusions of the manuscript. While it may seem the review comments are trivial, they are offered to enable to reader to more fully appreciate the innovation offered by the research.
Author Response
General response
We sincerely thank you for your unique perspective and recommendations in improving our manuscript. We appreciate your constructive comments on better describing the situation and need in LMIC, properly differentiating the CPAP, BiPAP and bubble-based devices and the suggestion for adding a bill of materials amongst other comments. Find below comments and corresponding revisions addressing raised concerns. This markedly improves the quality of this manuscript.
Comment 1
The rich content of this manuscript identifies a gap in non-invasive ventilation for the general category of infants beyond the neonatal period who may also be considered small children or toddlers. The research is very clearly outlined and the technology development, testing, and discussion is quite complete without any obvious deficiencies.
[Line 22-23] The reader is potentially left “hanging” on exactly the issue in the sentence “….but is insufficient for sicker children” Because? Due to? This, because the following sentence implies a solution to a problem that is ambiguous.
Response 1
Thank you for this comment on need for clarification in the abstract. This has now been addressed concisely and further explained in the introduction.
Comment 2
[Line 56-57] The use of the reference seems to be more to emphasize the LMIC need versus explain the unique aspects of the bubble CPAP. Even just adding a sentence that with bCPAP bubbles formed in water create pressure oscillations to prevent alveolar collapse. As the reader moves forward, the distinction between the common CPAP and BiPap in the previous paragraph (lines 47-48) is lost in translation to the bCPAP comments. While not a big deal, you are eventually comparing and contrasting CPAP, BiPAP, bCPAP, and now bi-level-bCPAP and the reader is required to read and re-read to understand the nuances between them. A table to compare and contrast might be helpful.
Response 2
These are excellent points. We have now added a description on how bubble CPAP works in the introduction in lines 61-65 and also addressed the description of the different modes.
Comment 3
[Lines 68-80] Excellent overall narrative leading to identification of the gap in technology, but is the issue about LMIC device cost, functionality, or both? It seems the current work offers the clinician a pathway where cost and functionality are not mutually exclusive. Perhaps the authors could consider adding a sentence that underscores the significance of the work in a more compelling way, such as: “There is no widely available, affordable, and effective bubble-based bilevel NIV system for infants beyond the neonatal stage and children under 5, despite the high burden of respiratory illness in this age group and the limitations of current technologies in LMICs.” If such a statement is true about the manuscript, it might help the reader for this type of direct statement to be added.
Response 3
Thank you very much for this great suggestion which is indeed true we have now incorporated the proposed statement in the introduction to improve the description of the gaps in LMICs.
Comment 4
[Lines 98-102] Given the idea the device would meet the needs of LMIC, it may help to have in an appendix a summary of the bill-of-materials and cost comparison to traditional devices to put “low cost” in context. Indeed, the manuscript is centered on a pilot trial of an experimental set-up, but the pathway or motivation to translation would be enhanced. It is unclear if the need to have high-end technology “….using custom 3D-printed parts and pulleys…” (Line 102) would be required at the LMIC site.
Response 4
Thank you for this suggestion. A table of bill of materials has been added and now appears in the appendix.
Comment 5
The methods, procedures, results and statistics meet contemporary standards for clarity and quality. And the results do support the conclusions of the manuscript. While it may seem the review comments are trivial, they are offered to enable to reader to more fully appreciate the innovation offered by the research.
Response 5
Thank you for this supportive comment. It is very much appreciated by the authors.

Round 2
Reviewer 1 Report
Comments and Suggestions for Authors
Authors have addressed most of the comments raised.